# Transcriptomic Analysis Reveals That Rho GTPases Regulate Trap Development and Lifestyle Transition of the Nematode-Trapping Fungus *Arthrobotrys oligospora*

Le Yang,[a,b,c] Xuemei Li,[a,b,c] Na Bai,[a,b,c] Xuewei Yang,[a,b,c] Ke-Qin Zhang,[a,b,c] Jinkui Yang[a,b,c]

[a]State Key Laboratory for Conservation and Utilization of Bio-Resources, Yunnan University, Kunming, People's Republic of China
[b]Key Laboratory for Microbial Resources of the Ministry of Education, Yunnan University, Kunming, People's Republic of China
[c]School of Life Sciences, Yunnan University, Kunming, People's Republic of China

**ABSTRACT** Nematode-trapping (NT) fungi can form unique infection structures (traps) to capture and kill free-living nematodes and, thus, can play a potential role in the biocontrol of nematodes. *Arthrobotrys oligospora* is a representative species of NT fungi. Here, we performed a time course transcriptome sequencing (RNA-seq) analysis of transcriptomes to understand the global gene expression levels of *A. oligospora* during trap formation and predation. We identified 5,752 unique differentially expressed genes, among which the *rac* gene was significantly upregulated. Alternative splicing events occurred in 2,012 genes, including the *rac* and *rho2* gene. Furthermore, we characterized three Rho GTPases (Rho2, Rac, and Cdc42) in *A. oligospora* using gene disruption and multiphenotypic analysis. Our analyses showed that AoRac and AoCdc42 play an important role in mycelium growth, lipid accumulation, DNA damage, sporulation, trap formation, pathogenicity, and stress response in *A. oligospora*. AoCdc42 and AoRac specifically interacted with components of the Nox complex, thus regulating the production of reactive oxygen species. Moreover, the transcript levels of several genes associated with protein kinase A, mitogen-activated protein kinase, and p21-activated kinase were also altered in the mutants, suggesting that Rho GTPases might function upstream from these kinases. This study highlights the important role of Rho GTPases in *A. oligospora* and provides insights into the regulatory mechanisms of signaling pathways in the trap morphogenesis and lifestyle transition of NT fungi.

**IMPORTANCE** Nematode-trapping (NT) fungi are widely distributed in terrestrial and aquatic ecosystems. Their broad adaptability and flexible lifestyles make them ideal agents for controlling pathogenic nematodes. *Arthrobotrys oligospora* is a model species employed for understanding the interaction between fungi and nematodes. Here, we revealed that alternative splicing events play a crucial role in the trap development and lifestyle transition in *A. oligospora*. Furthermore, Rho GTPases exert differential effects on the growth, development, and pathogenicity of *A. oligospora*. In particular, AoRac is required for sporulation and trap morphogenesis. In addition, our analysis showed that Rho GTPases regulate the production of reactive oxygen species and function upstream from several kinases. Collectively, these results expand our understanding of gene expression and alternative splicing events in *A. oligospora* and the important roles of Rho GTPases in NT fungi, thereby providing a foundation for exploring their potential application in the biocontrol of pathogenic nematodes.

**KEYWORDS** *Arthrobotrys oligospora*, alternative splicing, Rho GTPases, trap formation, lifestyle transition

Address correspondence to Ke-Qin Zhang, kqzhang1@ynu.edu.cn, or Jinkui Yang, jinkui960@ynu.edu.cn.

The authors declare no conflict of interest.

**N**ematophagous fungi are natural enemies of nematodes that predate on nematodes by developing trapping devices (traps) by producing extracellular enzymes and toxic compounds (1, 2). Nematode-trapping (NT) fungi, the main group of nematophagous fungi, are capable of developing specific traps like adhesive networks, adhesive knobs, and constricting rings to capture nematodes and extract nutrients from them (3). The trap formation is an important indicator for NT fungi to transition from a saprophytic to a predacious lifestyle (4, 5). Recently, several NT fungi producing different kinds of traps have been sequenced to elucidate the molecular basis of the underlying regulatory mechanism of trap development and lifestyle transition in NT fungi (4, 6–8). *Arthrobotrys oligospora* is a typical NT fungus that forms adhesive networks to capture nematodes and has been widely studied to understand the mechanisms of trap formation (9). Since the genome of *A. oligospora* was sequenced (4), several signaling proteins involved in the regulation of trap formation and pathogenicity, such as mitogen-activated protein kinase (MAPK) (10), G protein $\beta$ subunit (11), regulators of G protein signaling (12), and small GTPases Rab-7A (13) and Ras family (14), have been identified. These studies demonstrate the complexity of the cellular processes involved in trap formation and lifestyle transition, regulated by various signaling proteins and pathways in NT fungi.

Transcriptome sequencing (RNA-seq) technology has been widely used to elucidate the interactions between pathogenic fungi and hosts, such as *Magnaporthe oryzae* with *Oryza sativa* (15) and *Beauveria bassiana* with *Galleria mellonella* (16). Recently, Dong et al. (16) demonstrated an overview of alternative splicing (AS) events for *B. bassiana* growing in an insect hemocoel. AS, which generates different mature mRNAs from a single precursor mRNA, regulates various biological processes in an organism. Although the functionality of AS in fungi has not been explored fully, a substantive number of AS events have been reported in zygomycetes (17), basidiomycetes (18), and ascomycetes (19, 20). AS significantly contributes to the lifestyle transition and virulence of human and plant pathogens (e.g., *Cryptococcus neoformans* and *Magnaporthe grisea*) and affects genes associated with fungal stress responses and differentiation (21). However, little is known about AS events in NT fungi.

The Rho protein family, first identified in *Aplysia* species, belongs to the family of small GTPases (22, 23). Most Rho proteins are GTP-hydrolyzing molecular switches that participate in different signaling pathways for cell growth, morphogenesis, and gene regulation (24, 25). Several Rho proteins have been characterized in fungi, including Rho1, Rho2, Cdc42, and Rac1 (26, 27). Rho2 is highly homologous to Rho1 and plays a partially overlapping function in actin cytoskeleton repolarization and cell wall biosynthesis; however, it is of minor importance for growth and spore formation (28–31). Rac, highly homologous to Cdc42, has overlapping and distinct functions in dimorphic and filamentous fungi (32, 33). Cdc42 and Rac play altered important roles in different fungi. For example, in *Talaromyces* (formerly *Penicillium*) *marneffei* and *Candida albicans*, Rac1 is not essential for viability but is important for filamentous growth (34–36), whereas Cdc42 is a master regulator of cell polarity and is essential for viability in *T. marneffei* and *C. albicans* (35–38). In *Neurospora crassa*, *Aspergillus fumigatus*, and *Aspergillus niger*, Rac is essential for hyphal morphogenesis and apical branching, whereas Cdc42 plays a central role in hyphal morphogenesis in *Aspergillus nidulans* (29, 39–41). In addition, in *A. fumigatus* and *A. nidulans*, the knockout of the *rac* gene results in reduced production of reactive oxygen species (ROS) (40–42). Taken together, these studies show that the Rho family contains multifunctional proteins that play vital roles in fungal growth, cell polarity, cell wall biosynthesis, and ROS production. However, the roles of the Rho family proteins in NT fungi remain elusive.

In this study, we performed a time course RNA-seq analysis to probe the global gene expression levels during induction of *A. oligospora* with *Caenorhabditis elegans* (a model nematode). Taking the sample obtained at 0 h as a control, the differentially expressed genes (DEGs) and AS events were analyzed during the trap formation and nematode predation (12 to 48 h). Furthermore, the characterization of three homologous Rho

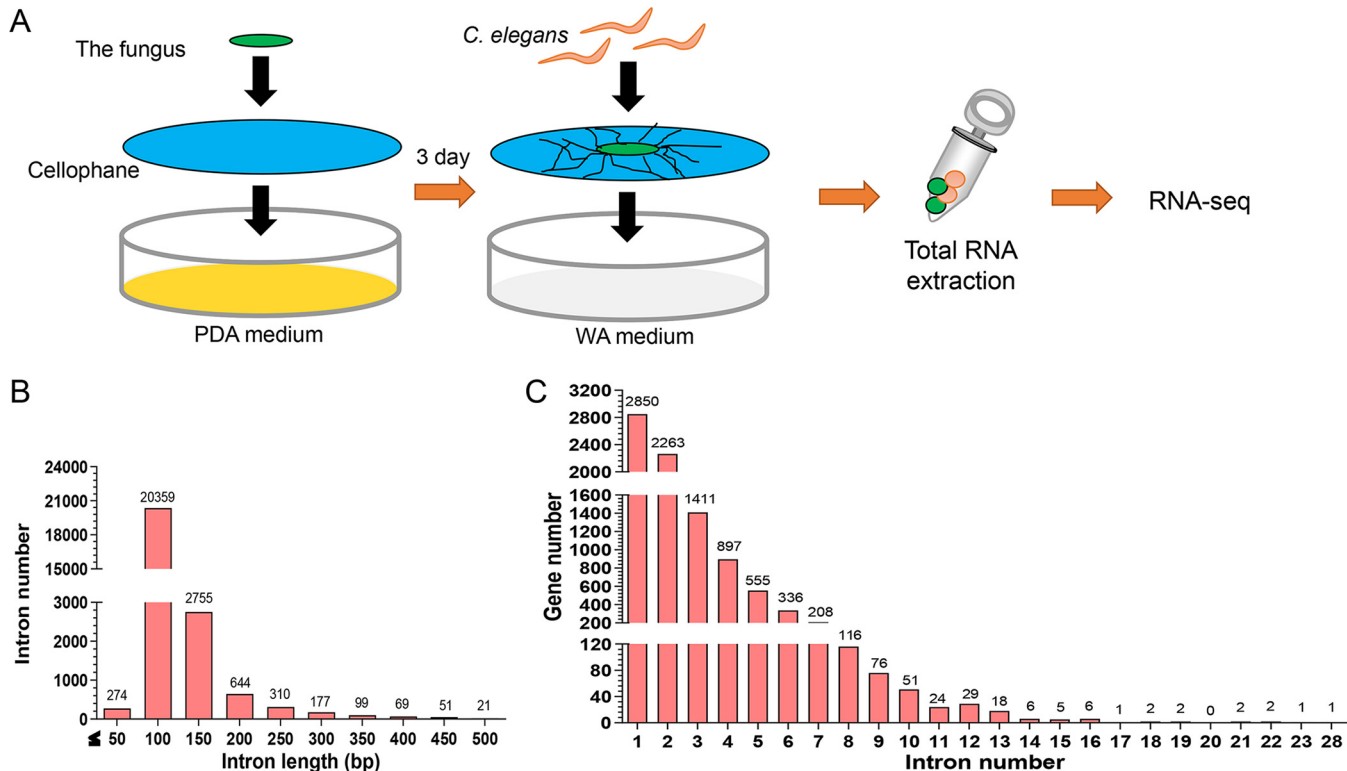

**FIG 1** Deciphering the transcriptome of *A. oligospora* during trap formation and nematode predation using the dual RNA-seq method. (A) *A. oligospora* hyphal blocks of the same size were inoculated into PDA medium covered with cellophane for 3 days and then transferred to water agar (WA) medium covered with cellophane. Approximately 5,000 nematodes were added to induce trap formation. Mixed samples were collected at 0, 12, 24, 36, and 48 h after adding nematodes (*C. elegans*), and three samples at each time point were used for RNA-seq. (B and C) Structural features of *A. oligospora* genes. Statistics of the length distribution of all introns (B) and intron composition of each gene (C) are shown.

proteins (Rho2 of *A. oligospora* [AoRho2], AoCdc42, and AoRac) through gene disruption and multiphenotypic comparison demonstrated the indispensable roles of Rho family proteins, especially AoRac and AoCdc42, in the growth, development, and pathogenicity of *A. oligospora*.

## RESULTS

**Transcriptome analysis of *A. oligospora* induced with nematode *C. elegans*.** To obtain real-time information, we used RNA-seq technology to sequence the transcriptome profiles of *A. oligospora* induced with nematode *C. elegans* at different time points (0 h, 12 h, 24 h, 36 h, and 48 h) (Fig. 1A). After sequencing a mixed sample (*A. oligospora* and *C. elegans*), fungal reads were extracted by searching the RNA-seq reads in the *A. oligospora* genome (4). Highly reproducible RNA-seq data were obtained from three biological replicates per culturing condition, with Spearman correlation coefficients above 0.92 for each sample (Fig. S1A in the supplemental material). A range of clear reads (5.99 to 9.04 Gb) was obtained per sample, and 83.9% to 97.1% of the reads were mapped to the *A. oligospora* genome (11,479 genes) (Table S1). Furthermore, 29 genes associated with MAPK signaling pathways and autophagy were selected to verify the transcriptomic data by real-time quantitative reverse transcription-PCR (RT-qPCR). The results revealed similar trends of transcriptomes and mRNA expression levels, indicating a relatively high consistency between the RNA-seq and RT-qPCR analyses (Table S2). Screening of the shared genes between all the samples identified 9,162 unique expressed genes, accounting for 79.8% of the annotated genes in the *A. oligospora* genome (Table S3). A survey of the *A. oligospora* genome revealed that 77.2% of the protein-coding genes ($n$ = 8,862) contained at least one intron (Fig. 1B and C), predicting an important layer of genome regulation conducted by AS. The 24,895 introns covered 5.2% of the genome sequence (40.07 Mb). The average intron length was 89 bp, and the intron-containing genes

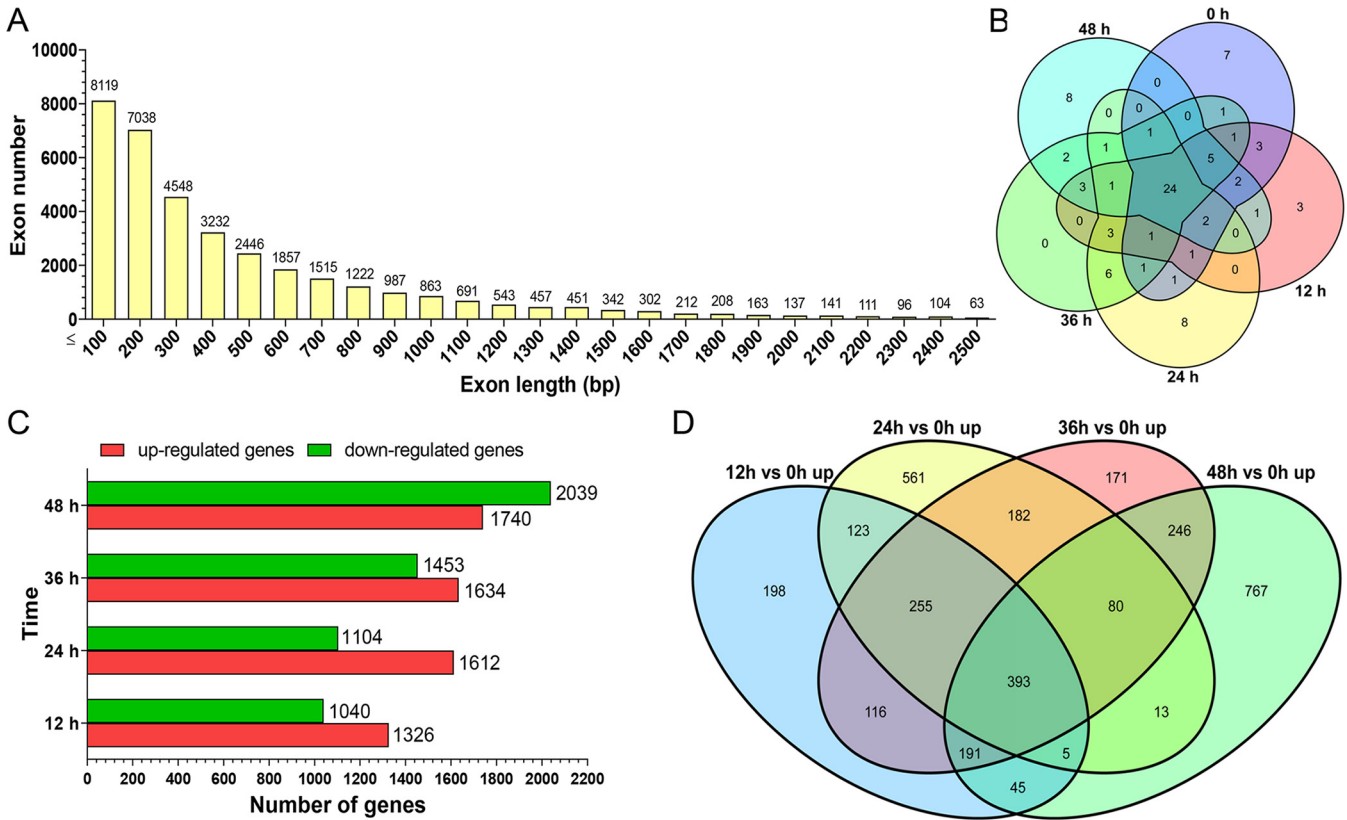

**FIG 2** Whole transcriptome of *A. oligospora* induced with nematodes. (A) Statistics of the length distributions of all exons. (B) Venn diagram analysis of the 50 most abundant genes of samples at each time point and 24 genes present at each time point. (C) Taking the 0-h sample as a control, the numbers of differentially expressed genes (DEGs) were analyzed during the trap formation and nematode predation (12 to 48 h). (D) Venn diagram analysis of the DEGs that were upregulated ($n = 393$) at all four time points.

harbored an average of 2.1 introns. A total of 36,375 annotated exons with an overall average length of 472 bp were identified. The exon-containing genes harbored an average of 3.1 exons (Fig. 2A).

Gene expression levels were normalized by the library and locus size and are indicated as reads per kilobase per million mapped reads (RPKM) (43). Venn diagram analysis of the 50 most abundant genes of the samples at each time point (RPKM values ranging from 63,116.0 to 1,337.2) identified 24 genes that were present at each time point (Fig. 2B). These 24 genes encoded eight hypothetical proteins, four heat shock proteins, two ATPase-related proteins, and 10 proteins representative of ubiquitin family protein, TRX family protein, eukaryotic initiation factor 1, and proteolipid membrane potential modulator (Table S4). These results suggest that abundant transcripts are primarily involved in protein modification and stress responses.

**Analysis of DEGs during the trap formation and nematode predation.** Taking the 0-h sample as a control, during the trap formation and nematode predation (12 to 48 h), the number of upregulated DEGs was higher than the number of downregulated DEGs in the samples at 12, 24, and 36 h, whereas the number of downregulated DEGs was increased at 48 h (Fig. 2C). After removing the DEGs that were repeated during the trap formation and nematode predation, we identified 5,752 unique DEGs (covering 50.1% of the genome). As trap formation is a dynamic developmental process, we focused on the upregulated DEGs. The Venn diagram analysis identified 393 DEGs upregulated at all four time points (Fig. 2D). Among these 393 DEGs, 73 DEGs were enriched ($P < 0.05$) in nine Kyoto Encyclopedia of Genes and Genomes (KEGG) pathways, including carbohydrate metabolism, amino acid metabolism, and lipid metabolism (Fig. 3A; Table S5). These results show that trap formation is an energy-consuming process. In addition, we also performed the KEGG enrichment analysis of the upregulated DEGs at

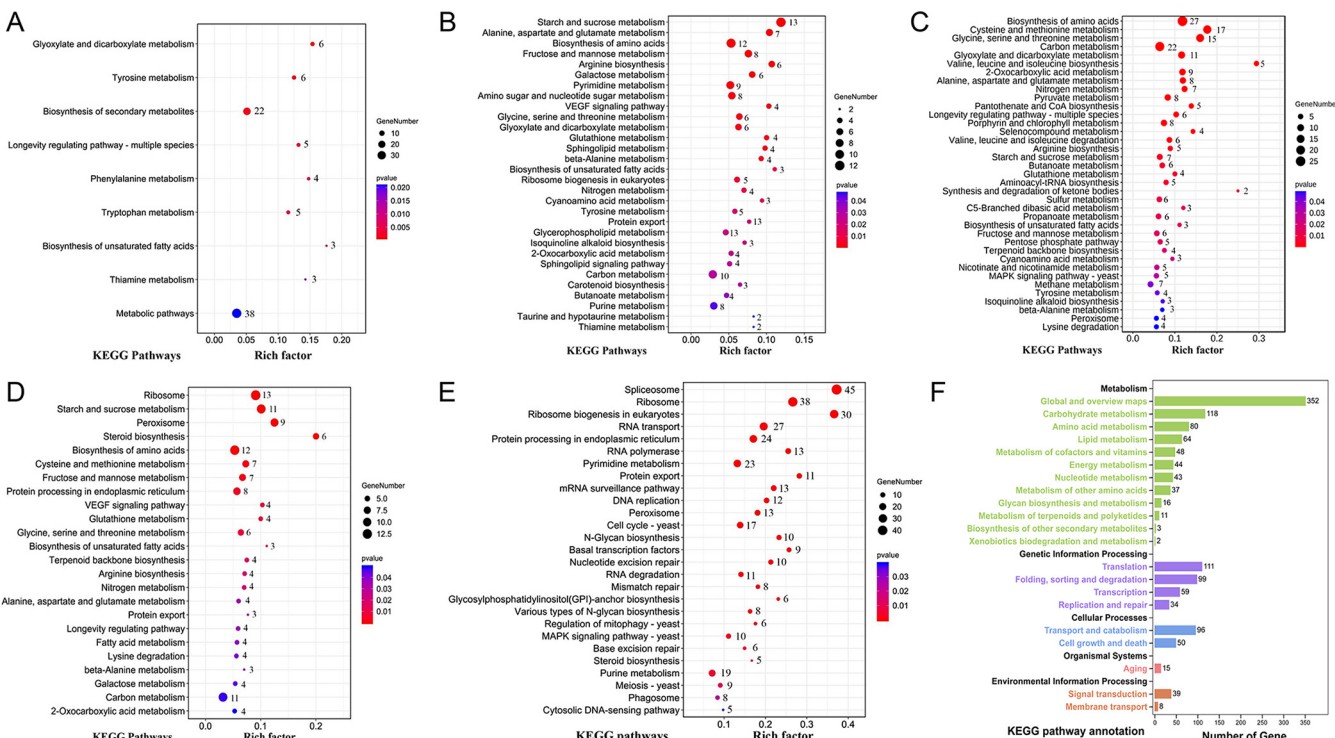

**FIG 3** Analysis of the DEGs and identification of AS events during the process of infection. (A) The 393 DEGs upregulated at all four time points were significantly enriched (*P* < 0.05) in nine Kyoto Encyclopedia of Genes and Genomes (KEGG) pathways. (B) KEGG enrichment of DEGs at 12 h: 1,326 DEGs were significantly enriched (*P* < 0.05) in 30 KEGG pathways. (C) KEGG enrichment of DEGs at 24 h: 1,612 DEGs were significantly enriched (*P* < 0.05) in 37 KEGG pathways. (D) KEGG enrichment of DEGs at 36 h: 1,634 DEGs were significantly enriched (*P* < 0.05) in 24 KEGG pathways. (E) KEGG enrichment of DEGs at 48 h: 1,740 DEGs were significantly enriched (*P* < 0.05) in 27 KEGG pathways. (F) Genes demonstrating alternative splicing (AS) were subjected to KEGG annotation and classified into 21 KEGG secondary classes.

each time point. At 12 h, 1,326 DEGs were enriched (*P* < 0.05) in 30 KEGG pathways (Fig. 3B; Table S5). Among these, 26 pathways belong to the metabolic category, including amino acid metabolism, carbon metabolism, and lipid metabolism. The remaining four pathways were related to ribosome biogenesis in eukaryotes, protein export, sphingolipid signaling, and vascular endothelial growth factor (VEGF) signaling. At 24 h, 1,612 DEGs were enriched (*P* < 0.05) in 37 KEGG pathways (Fig. 3C; Table S5), of which 33 pathways were related to metabolism and biosynthesis. The remaining four pathways were related to aminoacyl-tRNA biosynthesis, the longevity regulating pathway (LRP), the MAPK signaling pathway, and peroxisome. At 36 h, 1,634 DEGs (*P* < 0.05) enriched in 24 KEGG pathways were identified (Fig. 3D; Table S5). Eighteen of these pathways were related to metabolism and biosynthesis, three belonged to genetic information processing (ribosome, protein export, and protein processing in endoplasmic reticulum), and three were related to LRP, VEGF signaling, and peroxisome. At 48 h, 1,740 DEGs were enriched (*P* < 0.05) in 27 KEGG pathways (Fig. 3E; Table S5) that included 14 pathways related to genetic information processing (spliceosome, RNA transport, protein export, and basal transcription factors, etc.), 6 pathways related to metabolism, 5 pathways related to cellular processes (peroxisome, cell cycle, and phagosome, etc.), and 2 pathways related to MAPK signaling and cytosolic DNA sensing. Overall, the DEGs of *rac* were upregulated at three of the four time points (12, 36, and 48 h) and were involved in sphingolipid, VEGF signaling, and phagosome pathways. Rac is a small GTPase of the Rho family that regulates the polarization, hyphal growth, hyphal morphogenesis, conidial production, and pathogenicity of pathogenic fungi (44).

**Identification of AS events during the trap formation and nematode predation.** AS contributes significantly to the lifestyle transition and virulence of fungi and exerts significant effects on genes associated with fungal stress responses and differentiation (21). Transcriptome alignment identified 3,058 to 4,488 splicing events in 15 sample

**TABLE 1** Partial sequence properties of three Rho GTPases in *A. oligospora*

| Gene | Size of open reading frame (bp) | No. of: | | Isoelectric point | Mol wt (kDa) | G box motifs[a] |
| | | Introns | Amino acid residues | | | |
|---|---|---|---|---|---|---|
| *Aorho2* | 1,067 | 6 | 201 | 5.57 | 22.46 | G1 to G5 |
| *Aocdc42* | 804 | 3 | 195 | 5.47 | 21.61 | G1 to G5 |
| *Aorac* | 1,079 | 4 | 264 | 8.40 | 29.10 | G1 to G5 |

[a]Conserved motifs are marked in Fig. S2.

libraries (Fig. S2A; Table S6). All splicing events were categorized into five types: retention intron (RI), alternative 5′ splice site (A5SS), alternative 3′ splice site (A3SS), mutually exclusive exon (MXE), and skipped exon (SE). The most abundant AS type was SE, accounting for ~37.2% of splicing events, followed by RI (29.7%), A3SS (19.0%), A5SS (11.2%), and MXE (2.7%). Relative to 0 h, during the trap formation and nematode predation, the transcriptome alignment identified 3,430, 3,489, 3,764, and 3,748 AS events at 12, 24, 36, and 48 h, respectively (Table S6). The AS events in the 12- to 48-h libraries occurred in 1,629, 1,677, 1,729, and 1,754 genes, representing 14.1%, 14.6%, 15.0%, and 15.2% of total genes, respectively. Overlapping genes between the three replicates were considered final results for each splicing type. Overall, AS events occurred in 2,012 genes during trap formation and nematode predation. To understand the functionality of the genes demonstrating AS, they were subjected to KEGG annotation and enrichment analysis. The analysis classified these genes into 21 KEGG secondary classes, including the metabolism of carbohydrates, amino acids, energy, and lipids, transcription, reproduction and translation of genetic information, transport, and signal transduction. Among them, seven pathways were significantly enriched ($P < 0.05$), including the tricarboxylic acid (TCA) cycle, metabolism, RNA transport, and phagosome (Fig. 3F; Table S7). To further understand the involvement of AS in fungal pathogenicity, 737 genes potentially involved in pathogenicity were identified via BLAST analysis of the AS-demonstrating genes against the PHI database (Table S8). Of these, 482 genes were annotated as loss or reduction of pathogenicity and lethal. These genes, including signal transduction-related genes (e.g., *ste50*, *hog1*, and *slt2*), autophagy or mitophagy-related genes (e.g., *atg4*, *atg15*, and *mtor*), small GTPases (e.g., *rab7*, *rac*, and *rho2*), and several other genes (e.g., *hog1*, *slt2*, *atg4*, and *rab7*), were confirmed to contribute to *A. oligospora* development and pathogenicity (13, 14, 45, 46). Rho2 and Rac, the small GTPase of the Rho family, regulate cell wall integrity in filamentous fungi (44). Furthermore, RT-PCR analysis confirmed the AS events of *Aorho2* and *Aorac* genes (Fig. S2B and C). These results suggested a possible important role of Rho GTPases in the trap formation and nematode predation of *A. oligospora*. Therefore, the three Rho proteins Rho2, Rac, and Cdc42, closely related to Rac, were selected to validate the effects of Rho GTPases on the mycelial development, differentiation, and pathogenicity of *A. oligospora*.

**Sequence and transcription analyses of three Rho GTPases.** The partial sequence properties of AoRho2, AoCdc42, and AoRac of *A. oligospora* are summarized in Table 1 and Fig. S3. InterProScan analysis showed that these proteins have no signal peptide, but all shared a conserved P loop containing nucleoside triphosphate hydrolases (IPR027417) and a small GTP-binding protein domain (IPR005225). The predicted three-dimensional (3-D) structures of AoRho2, AoCdc42, and AoRac proteins were not similar, suggesting that they might be involved in different functions in *A. oligospora* (Fig. S4A). A phylogenetic tree of Rho GTPases from diverse filamentous fungi was constructed based on their amino acid sequences, and the orthologs of Rho2, Cdc42, and Rac were separated into three clades (Fig. S4B). Furthermore, the increased transcription levels of *Aorho2*, *Aocdc42*, and *Aorac* during the incubation of *A. oligospora* with nematodes suggested the upregulation of these genes during trap formation (Fig. S2D).

**Aocdc42 and Aorac regulate mycelial growth and mycelial morphology.** To study the function of *Aorho2*, *Aocdc42*, and *Aorac*, three mutants of each Rho GTPase

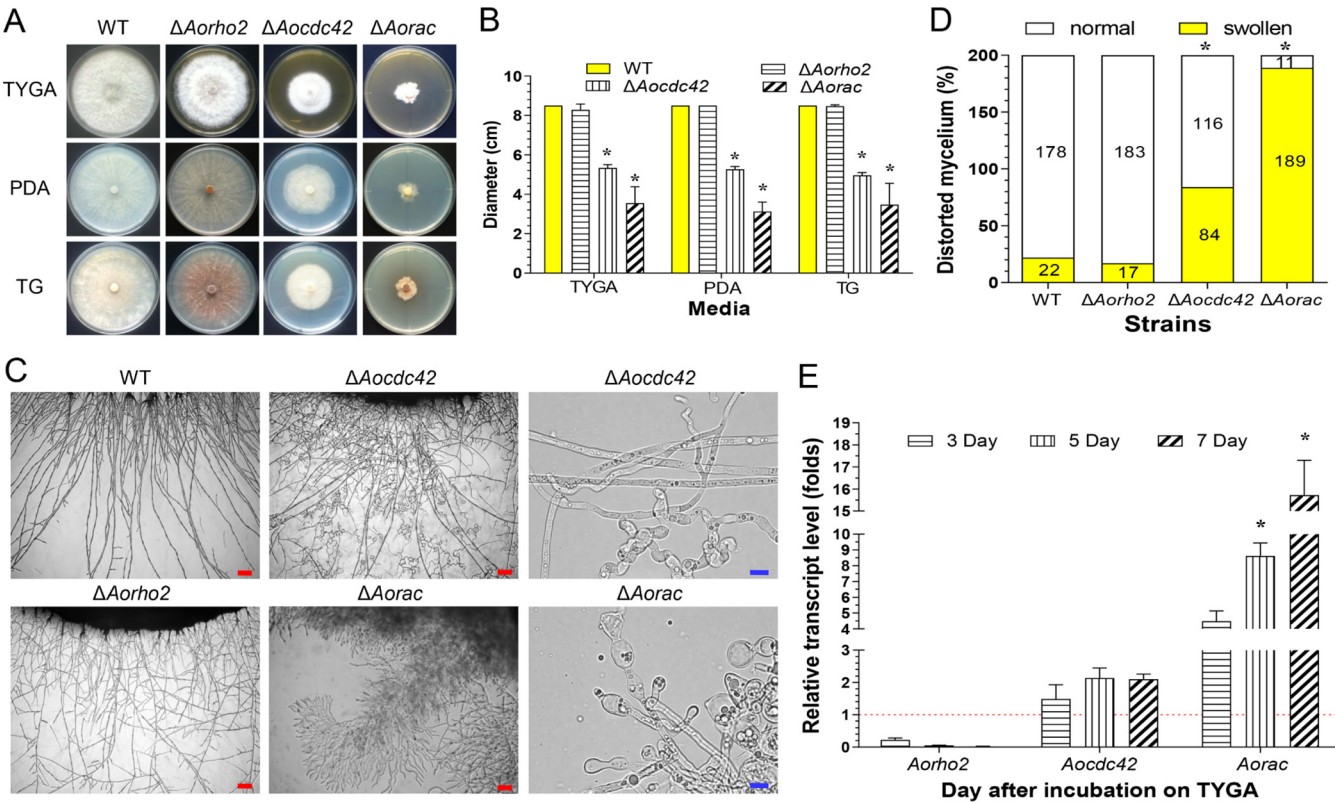

**FIG 4** Comparison of colony morphologies and mycelial growth between the wild-type (WT) and mutant (Δ*Aorho2*, Δ*Aocdc42*, and Δ*Aorac*) strains. (A) Colonies from the WT and mutant strains were cultured on PDA, TG, and TYGA plates for 7 days at 28°C. (B) Colony diameters of the WT and mutant strains cultured on different media for 7 days. (C) Hyphae of each strain grown on PDA plates for 3 days and monitored with a light microscope. Red bar = 100 $\mu$m, blue bar = 10 $\mu$m. (D) Proportions of deformed hyphal cells of the WT and mutant strains cultured on PDA media for 7 days. (E) Relative transcript levels (RTLs) of *Aorho2*, *Aocdc42*, and *Aorac* genes in the WT strain at different time points (fold changes). The red line indicates the standard (with RTL = 1) for statistical analysis of the RTL of each gene in the respective deletion mutant compared to that in the WT strain under a given condition. (B and E) Bars and error bars represent mean values ± SD. (B, D, and E) *, significant difference between mutant and WT strain ($n = 3$ for the WT strain [B and D], $n = 9$ for each mutant strain [B and D], and $n = 3$ for each gene [E]) (Tukey's HSD, $P < 0.05$).

gene were generated by replacing the open reading frame with the hygromycin resistance gene as a selectable marker in *A. oligospora*, and the generated gene deletion was confirmed by PCR and Southern blot analyses (Fig. S5). The hyphal growth and colony morphology of the wild type (WT) and each mutant strain were compared on different media, including tryptone-yeast extract-glucose agar (TYGA), potato dextrose agar (PDA), and tryptone-glucose (TG). Knockout of *Aocdc42* and *Aorac* resulted in slow growth in all three media, and knockout of *Aorho2* showed no significant change compared with the growth of the WT (Fig. 4A and B). The average colony diameters of the WT and the Δ*Aorho2*, Δ*Aocdc42*, and Δ*Aorac* mutant strains on the TYGA medium were 8.5, 8.3, 5.3, and 3.5 cm, respectively, those on the PDA medium were 8.5, 8.5, 5.3, and 3.1 cm, and those on the TG medium were 8.5, 8.5, 5.0, and 3.5 cm (Fig. 4B). The colony morphologies of the Δ*Aorac* mutant strains were irregular, and the Δ*Aorho2* mutant strains produced brown pigment on the TG medium (Fig. 4A). Furthermore, substantial hyperbranching or twisting was observed in the Δ*Aorho2*, Δ*Aocdc42*, and Δ*Aorac* mutants, which was more pronounced in the Δ*Aocdc42* and Δ*Aorac* mutants. Hyphal swelling was also observed in the Δ*Aocdc42* and Δ*Aorac* mutants (Fig. 4C). This hyperbranching and swelling found in hyphae of the Δ*Aocdc42* and Δ*Aorac* mutant strains was not commonly observed in the other mutants or the WT strain. Moreover, compared with the WT, about 42% and 95% of the mycelia of the Δ*Aocdc42* and Δ*Aorac* mutants, respectively, were swollen (Fig. 4D). Hyphal tips of both Δ*Aocdc42* and Δ*Aorac* mutants showed meandering growth, in contrast to the straight hyphae of WT, indicating that both *Aocdc42* and *Aorac* were at least partially involved in the maintenance of hyphal polarity. The transcript expression levels of *Aocdc42* and *Aorac* were

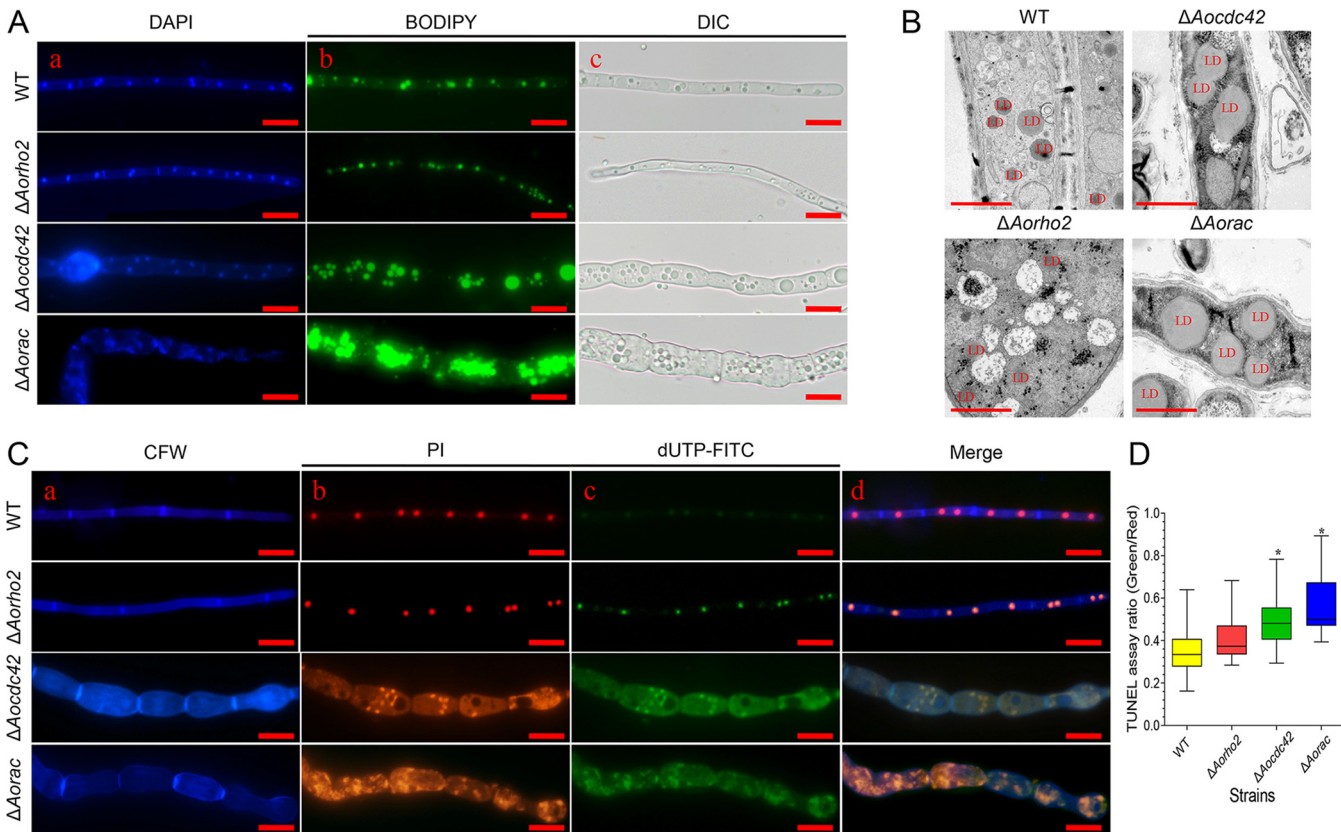

**FIG 5** Comparison of lipid accumulation, DNA fragmentation, and cell apoptosis in mycelia of WT and mutant strains. (A) Mycelia of the WT and mutants stained with DAPI and the fluorescent dye boron dipyrromethene (BODIPY). (a) DAPI, (b) BODIPY, (c) observed under differential interference contrast microscopy (DIC). Samples were examined under a confocal laser scanning microscope. Bar = 10 $\mu$m. (B) Lipid droplets (LDs) observed in the WT and mutant using transmission electron microscopy. Bar = 2 $\mu$m. (C) Mycelial morphology and TUNEL assay. Fungi were cultured on the PDA medium for 7 days, and mycelia were then stained with calcofluor white (CFW), followed by TUNEL analysis. (a) Mycelia stained with CFW. (b) Nuclei of the hyphae stained with PI. (c) Free DNA of the hyphae restained with FITC-dUTP. (d) Merged images from panels a to c. Samples were examined under a confocal laser scanning microscope. Bar = 10 $\mu$m. (D) Analysis of DNA fragmentation and cell apoptosis in hyphal cells. The ratio of green to red fluorescence intensity was determined for at least 30 fields observed under a microscope. (D) Error bars show SD. *, significant difference between mutant and WT strain (Tukey's HSD, $P < 0.05$).

higher during the conidiation stage (from 3 to 7 days) than during the vegetative-growth stage (day 2). In particular, the expression of *Aorac* was remarkably upregulated during different time points of the conidiation stage (Fig. 4E).

**Deletion of *Aocdc42* and *Aorac* resulted in lipid accumulation and DNA damage in hyphae.** The lipids in mycelia were observed by fluorescent dye boron dipyrromethene (BODIPY) staining and transmission electron microscopy. Compared with the WT, large amounts of lipids were accumulated in the Δ*Aocdc42* and Δ*Aorac* mutant strains, especially in the swollen mycelia (Fig. 5A and B). After staining the cell nucleus with 4',6-diamidino-2-phenylindole (DAPI), multiple nuclei were observed in hyphal cells of the WT strain, whereas the nuclei of the Δ*Aocdc42* and Δ*Aorac* mutants were difficult to stain, as they appeared diffused and fragmented (Fig. 5A). A prominent feature of cell apoptosis is the degradation of chromosomal DNA, which is a relatively common phenomenon (47). Therefore, the fragmentations of chromosomal DNA in the mycelia of the WT and mutants were detected via terminal deoxynucleotidyl transferase-mediated dUTP nick end labeling (TUNEL) assay (48), and the nuclei were stained using propidium iodide (PI). Both the WT and mutant strains were stained with fluorescein isothiocyanate (FITC)-dUTP, and the nuclei of Δ*Aocdc42* and Δ*Aorac* mutants were observed to be in a diffused form. Moreover, the FITC-dUTP fluorescence intensity (FI) of the mutants was higher than that of the WT (Fig. 5C). As most of the nuclei of the mutants were distributed in a diffused state, the FI ratios of FITC-dUTP and PI were estimated to analyze the differences in DNA damage and cell apoptosis

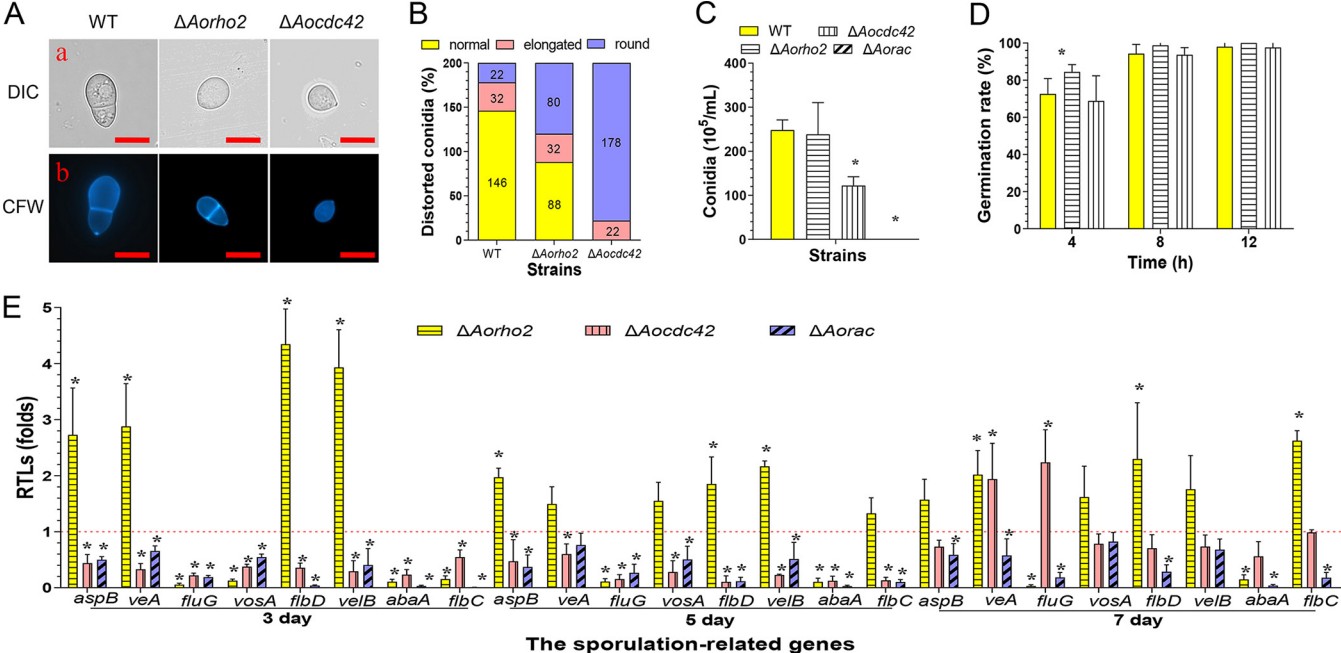

**FIG 6** Comparison of conidiation, conidial morphologies, and germination rates between the WT and mutant strains. (A) Conidia of the WT and mutants stained with CFW. (a) Conidial morphologies of the WT and mutant strains observed under DIC. (b) Conidia stained with CFW and observed under a confocal laser scanning microscope. Bar = 10 $\mu$m. (B) Proportions of deformed spores of the WT and mutant strains cultured on CMY medium for 14 days. (C) Comparison of conidial yields between the WT and mutant strains. (D) Conidial germination rates of the WT and mutants. (E) RTLs of sporulation-related genes in the mutants compared with the WT strain at different time points. The red line indicates the standard (RTL = 1) for statistical analysis of the RTL of each gene in the respective deletion mutant compared to that in the WT strain under a given condition. (C to E) Error bars show SD. *, significant difference between mutant and WT strain ($n = 3$ for the WT strain [C to E], $n = 9$ for each mutant strain [C to E], and $n = 3$ for each gene [F]) (Tukey's HSD, $P < 0.05$).

between the WT and mutants. The FI ratios of WT and $\Delta Aorho2$, $\Delta Aocdc42$, and $\Delta Aorac$ mutant strains were 0.35, 0.39, 0.48, and 0.56, respectively (Fig. 5D).

**Aocdc42 and Aorac regulate conidiation.** The disruption of the *Aocdc42* gene resulted in reduced conidial production, whereas the disruption of the *Aorac* gene led to loss of the ability to produce conidia. The spore yields of the WT and the $\Delta Aorho2$ and $\Delta Aocdc42$ mutants were estimated to be $2.5 \times 10^7$, $2.4 \times 10^7$, and $1.2 \times 10^7$ spores/ml, respectively (Fig. 6C). The conidia of the WT strain are usually obovoid in shape, with one septum formed near the base of the spore (49). In contrast, all conidia of the $\Delta Aocdc42$ mutant were morphologically abnormal—89% were smaller and round, and 11% were elongated—compared with those of the WT strain. Similarly, relative to the conidia of the WT strain, 40% of the conidia of the $\Delta Aorho2$ mutant were smaller and round, and 16% were elongated (Fig. 6A and B). However, compared with the WT, no difference in conidial germination of the $\Delta Aorho2$ and $\Delta Aocdc42$ mutants was observed. At 8 h, 94.3%, 98.8%, and 93.7% of the spores of the WT strain and the $\Delta Aorho2$ and $\Delta Aocdc42$ mutants, respectively, were germinated (Fig. 6D). The expression levels of eight sporulation-related genes, *abaA*, *aspB*, *flbC*, *flbD*, *fluG*, *veA*, *velB*, and *vosA*, were determined in the WT and mutants by RT-qPCR after they were cultured on corn-maizena yeast extract (CMY) medium for 3, 5, and 7 days, respectively. Our results showed that the expression levels of these eight genes were significantly downregulated ($P < 0.05$) at each time point in the $\Delta Aorac$ mutants compared with their expression levels in the WT, whereas their expression levels were significantly downregulated ($P < 0.05$) in $\Delta Aocdc42$ mutants on the 3rd and 5th days. Additionally, two genes (*fluG* and *veA*) were upregulated, and other genes did not show a significant difference on the 7th day. In contrast, in the $\Delta Aorho2$ mutant, four genes (*flbD*, *aspB*, *veA*, and *velB*) were upregulated on the 3rd day, and the expression levels of six genes (*aspB*, *veA*, *vosA*, *flbD*, *velB*, and *flbC*) were upregulated on the 5th and 7th days. In contrast, the expression levels of *abaA* and *fluG* were significantly downregulated ($P < 0.05$) at each time point (Fig. 6E).

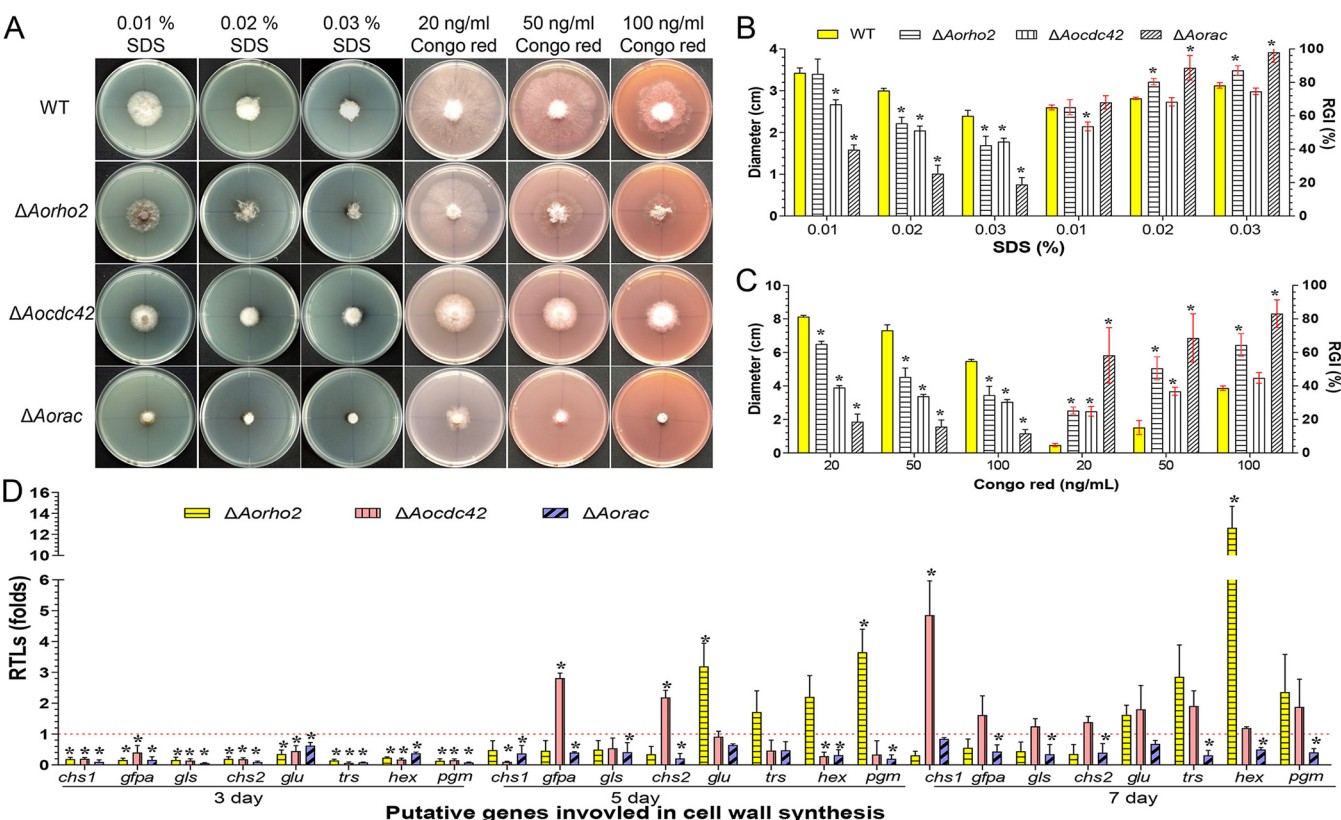

**FIG 7** Comparison of stress tolerance abilities of strains exposed to cell wall-perturbing agents. (A) Colony morphologies of the WT and mutant strains in the presence of cell wall-perturbing agents. (B and C) Colony diameters and relative growth inhibition (RGI) values of the strains cultured in the presence of 0.01 to 0.03% SDS (B) or 20 to 100 ng/ml Congo red (C). (D) RTLs of cell wall synthesis-related genes in the mutants compared with the WT strain at different time points. The red line indicates the standard (RTL = 1) for statistical analysis of the RTL of each gene in the respective deletion mutant compared to that in the WT strain under a given condition. (B to D) Error bars show SD. *, significant difference between mutant and WT strain ($n = 3$ for the WT strain [B and C], $n = 9$ for each mutant strain [B and C], and $n = 3$ for each gene [D]) (Tukey's HSD, $P < 0.05$).

*__Aorho2__, __Aocdc42__, and __Aorac__ regulate stress response.* Compared with the WT, the Δ*Aorac* mutants showed remarkable differences in sensitivity to chemical stressors. The relative growth inhibition (RGI) values of Δ*Aorac* mutants were significantly increased ($P < 0.05$) in the presence of any concentration of $H_2O_2$ (5 to 15 mM), menadione (0.01 to 0.05 mM), sodium dodecyl sulfate (SDS; 0.01 to 0.03%), Congo red (20 to 100 ng/ml), NaCl (0.1 to 0.3 M), or sorbitol (0.25 to 0.5 M). Similarly, the Δ*Aorho2* mutants showed differential sensitivities to these chemical agents. The RGI values of the Δ*Aorho2* mutants were significantly increased ($P < 0.05$) in the presence of 5 mM $H_2O_2$, menadione (0.03 to 0.05 mM), SDS (0.02 to 0.03%), Congo red (20 to 100 ng/ml), NaCl (0.1 to 0.3 M), or sorbitol (0.25 to 0.5 M). In contrast, the Δ*Aocdc42* mutants were only suppressed in Congo red (20 to 50 ng/ml) or sorbitol (0.25 to 0.5 M) (Fig. 7A, B, and C; Fig. S6A, B, and C and Fig. S7A and B).

To probe the regulatory mechanisms of Rho GTPases in stress resistance, the expression levels of seven putative genes involved in antioxidation, *gld*, *gstA*, *glr*, *prx*, *trxB*, *trxR*, and *trx*, were analyzed. The expression levels of these seven genes were downregulated in the Δ*Aorac* mutants at the tested time points. Compared with their expression levels in the WT, the expression levels of five genes, namely, *gld*, *gstA*, *glr*, *trxR*, and *trx*, were up-regulated in Δ*Aorho2* mutants on the 3rd day, one gene (*trx*) was upregulated on the 5th day, and five genes (*gld*, *gstA*, *glr*, *prx*, and *trx*) were upregulated on the 7th day. In the Δ*Aocdc42* mutants, the expression levels of five genes (*gld*, *glr*, *trxB*, *trxR*, and *trx*) were upregulated on the 3rd day, four genes (*gld*, *glr*, *trxR*, and *trx*) were upregulated on the 5th day, and one gene (*gld*) was upregulated on the 7th day (Fig. S6D).

Similarly, the expression levels of the *chs1*, *gfpa*, *gls*, *chs2*, *glu*, *hex*, and *pgm* genes, involved in cell wall synthesis, and *trs*, involved in trehalose synthesis, were compared

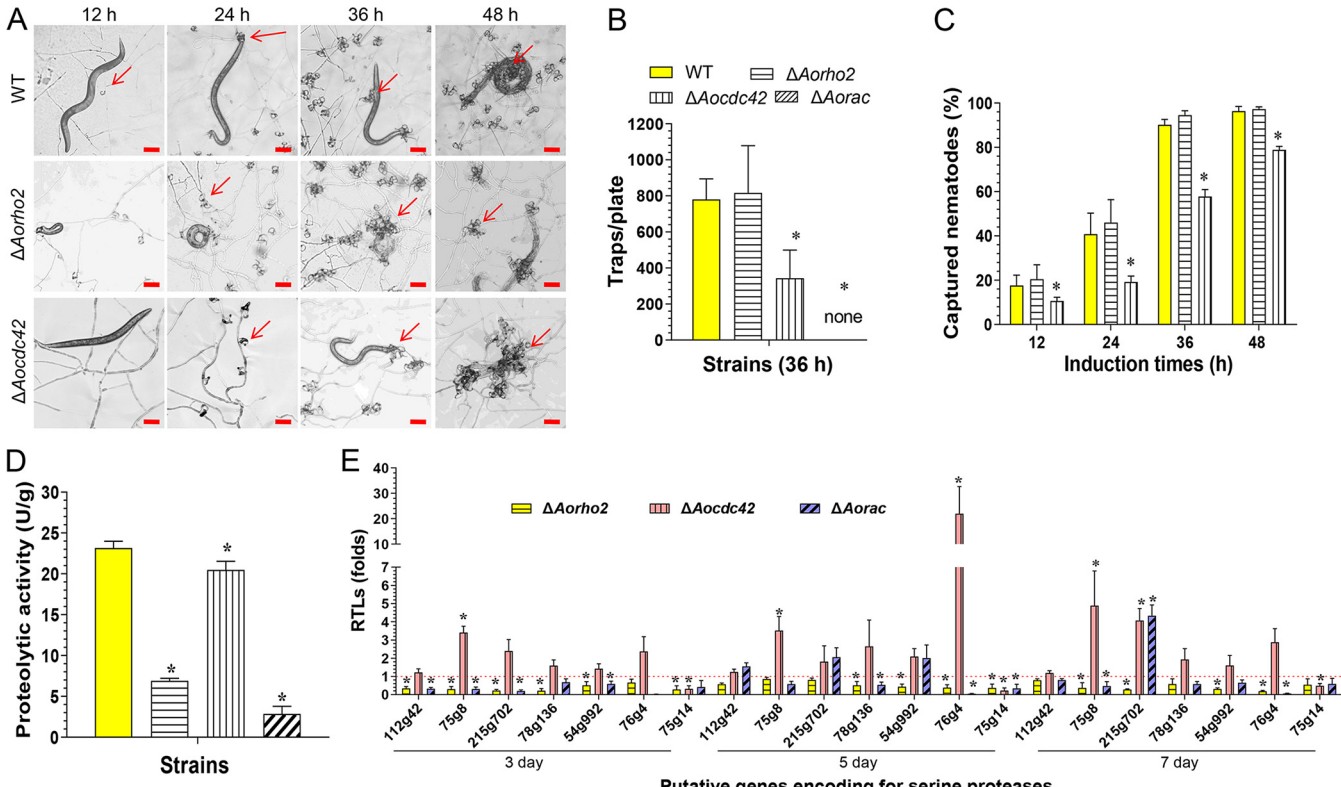

**FIG 8** Comparison of trap formation, nematocidal activities, and extracellular proteolytic activities in the WT and mutant strains. (A) Trap formation in the WT and mutant strains induced by nematodes at different time points. Red arrows show the traps produced by the WT strain and mutants. Bar = 50 $\mu$m. (B) Numbers of traps produced by the WT and mutant strains. (C) Percentages of captured nematodes at different time points. (D) Comparison of the total extracellular proteolytic activities of the WT and mutant strains exhibited on the 7-day-old PD broth. (E) RTLs of the genes encoding serine proteases in the mutants compared with the WT strain at different time points. The red line indicates the standard (RTL = 1) for statistical analysis of the RTL of each gene in the respective deletion mutant compared to that in the WT strain under a given condition. (B to E) Error bars show SD. *, significant difference between mutant and WT strain ($n$ = 3 for the WT strain [B to D], $n$ = 9 for each mutant strain [B to D], and $n$ = 3 for each gene [E]) (Tukey's HSD, $P$ < 0.05).

between the WT and mutant strains. The expression levels of these genes were down-regulated on the 3rd day. Additionally, the expression levels of these genes were downregulated in $\Delta$*Aorac* mutants on the 5th and 7th days. On the 5th day, two genes (*gfpa* and *chs2*) in $\Delta$*Aocdc42* mutants and four genes (*glu*, *trs*, *hex*, and *pgm*) in $\Delta$*Aorho2* mutants were upregulated. On the 7th day, all of these genes in $\Delta$*Aocdc42* mutants and four of these genes (*glu*, *trs*, *hex*, and *pgm*) in $\Delta$*Aorho2* mutants were up-regulated (Fig. 7D).

**Rho GTPases regulate the biocontrol potential of *A. oligospora*.** After the conidia of the WT and mutant strains ($\Delta$*Aorho2* and $\Delta$*Aocdc42*) were incubated on water agar (WA) plates at 28°C for 3 days, nematodes were added to induce the fungal strains to produce traps. The WT and $\Delta$*Aorho2* mutant strains produced immature traps containing one or two hyphal rings at 12 h, whereas the $\Delta$*Aocdc42* mutants did not produce traps. Subsequently, at 24 h and 36 h, the WT and $\Delta$*Aorho2* mutant strains formed mature traps containing five or more hyphal rings, and a few mature traps were produced by the $\Delta$*Aocdc42* mutants (Fig. 8A and B). Almost all of the nematodes were captured by the WT (90.2%) and $\Delta$*Aorho2* (94.5%) mutant strains, whereas only 57.7% of nematodes were captured by the $\Delta$*Aocdc42* mutants at 36 h (Fig. 8C). Since the $\Delta$*Aorac* mutants did not produce conidia, mycelium blocks were cultured in CMY medium at 28°C for 5 days, and then nematodes were added to induce trap formation. At 36 h, the WT strain produced a large number of traps and resulted in a higher killing rate of the nematodes, whereas the $\Delta$*Aorac* mutants did not produce traps until 48 h (Fig. S8A and B).

Moreover, the proteolytic activities of the $\Delta$*Aorho2* and $\Delta$*Aorac* mutants were significantly decreased on casein plates (Fig. S8C). Additionally, the proteolytic activity of

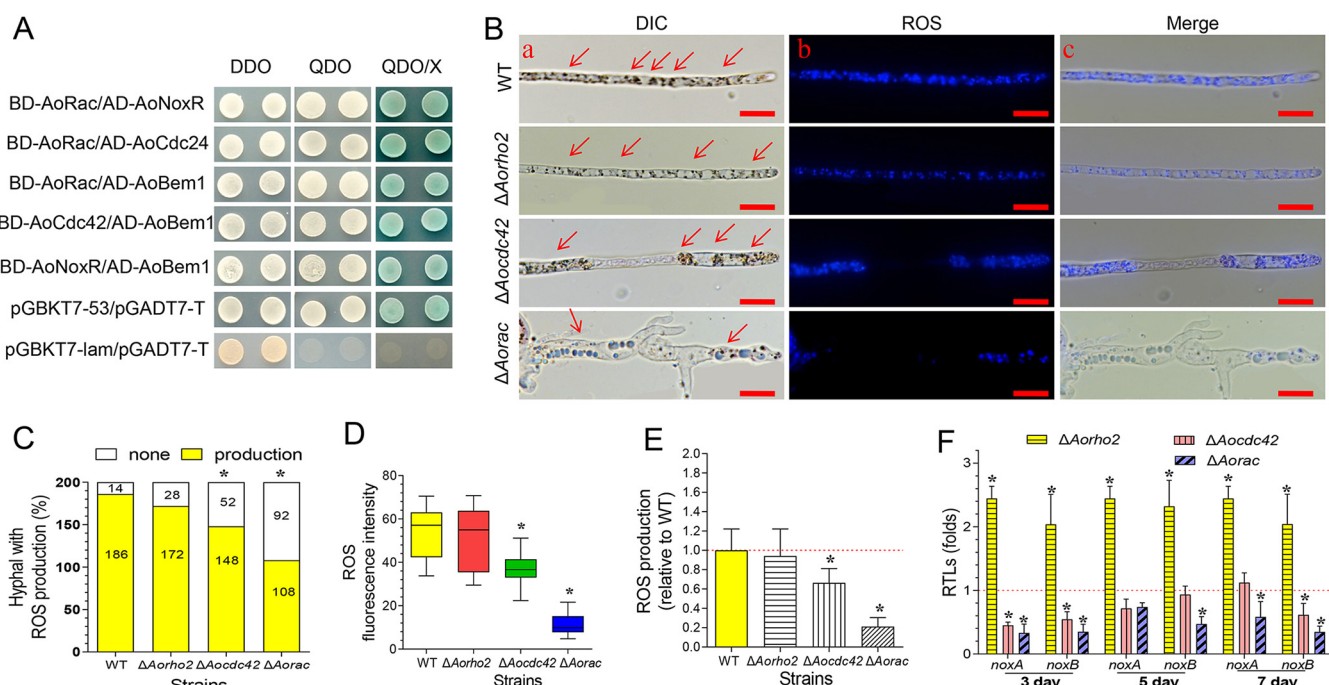

**FIG 9** Interactions of AoCdc42 and AoRac with components of Nox complex and comparison of ROS production of WT and mutant strains. (A) Yeast two-hybrid assay with AoCdc42 or AoRac as the bait and AoBem1, AoNoxR, and AoCdc24 as the prey. The interactions between pGBKT7-53 and pGADT7-T and between pGBKT7-lam and ADT7T were the positive and negative control, respectively. (B) Mycelia of the WT and mutants stained with dihydroethidium (DHE). (a) DIC showing the production of ROS; hyphal tips stained with DHE are shown by arrowheads. (b) DHE-stained hypha observed under a confocal laser scanning microscope. (c) Merged images from panels a and b. Red arrows indicate the ROS produced by the WT strain and mutants. Bar = 10 $\mu$m. (C) Frequencies of DHE-stained hyphae of WT and mutants. Numbers in the bars indicate hyphal counts. (D) Analysis of the ROS in hyphal cells. ROS production was detected as fluorescence intensity, which was determined for at least 30 fields observed under a microscope. (E) Values of fluorescence intensity relative to that of the WT. (F) RTLs of the Nox-encoding genes (*AonoxA* and *AonoxB*) in the mutants compared with the WT strain at different time points. The red line indicates the standard (with RTL = 1) for statistical analysis of the RTL of each gene in the respective deletion mutant compared to that in the WT strain under a given condition. (E and F) Bars and error bars represent mean values ± SD. (C to F) *, significant difference between mutant and WT strain (*n* = 3 for each gene [F]) (Tukey's HSD, *P* < 0.05).

the WT strain was 23.1 U/g hyphae, whereas those of Δ*Aorho2* and Δ*Aorac* mutants were decreased by 70.2% (6.9 U/g) and 87.9% (2.8 U/g) (Fig. 8D), respectively. In addition, the expression levels of seven genes (*112g42*, *75g8*, *215g702*, *78g136*, *54g992*, *76g4*, and *75g14*) encoding serine proteases were downregulated in the Δ*Aorho2* mutant at the tested time points. On the contrary, in the Δ*Aocdc42* mutant, the expression levels of six genes were upregulated, and one gene (*75g14*) was downregulated at the tested time points. In addition, the expression of the *PII* gene (*76g4*) was considerably upregulated (21.9-fold) compared with its expression in the WT on the 5th day. In the Δ*Aorac* mutant, the expression levels of all genes were downregulated on the 3rd day, three genes were upregulated and four were downregulated on the 5th day, and six genes were downregulated and one gene was upregulated on the 7th day (Fig. 8E).

**AoCdc42 and AoRac specifically interact with components of the Nox complex and regulate ROS production.** Previous studies have shown that Rac plays a conserved role in ROS production in mammals, plants, and fungi (50). To probe the role of Δ*Aocdc42* and Δ*Aorac* in ROS production of *A. oligospora*, we tested the possible interactions between AoCdc42 and AoRac and components of the Nox complex (AoNoxR, AoBem1, and AoCdc24) by performing yeast two-hybrid (Y2H) assays. The results demonstrated that AoNoxR, AoBem1, and AoRac interacted with each other, whereas AoCdc42 and AoRac interacted with AoBem1 and AoCdc24, respectively (Fig. 9A). Furthermore, we investigated the effects of *Aorho2*, *Aocdc42*, and *Aorac* deletions on hyphal ROS production. The hyphal production of ROS was compared between WT and mutant strains by counting dihydroethidium (DHE)-stained hyphae. The percentages of DHE-stained hyphae in the WT, Δ*Aorho2*, Δ*Aocdc42*, and Δ*Aorac* strains were 93%, 86%, 74%, and 54%, respectively (Fig. 9B and C; Fig. S8D). The FI values for DHE in

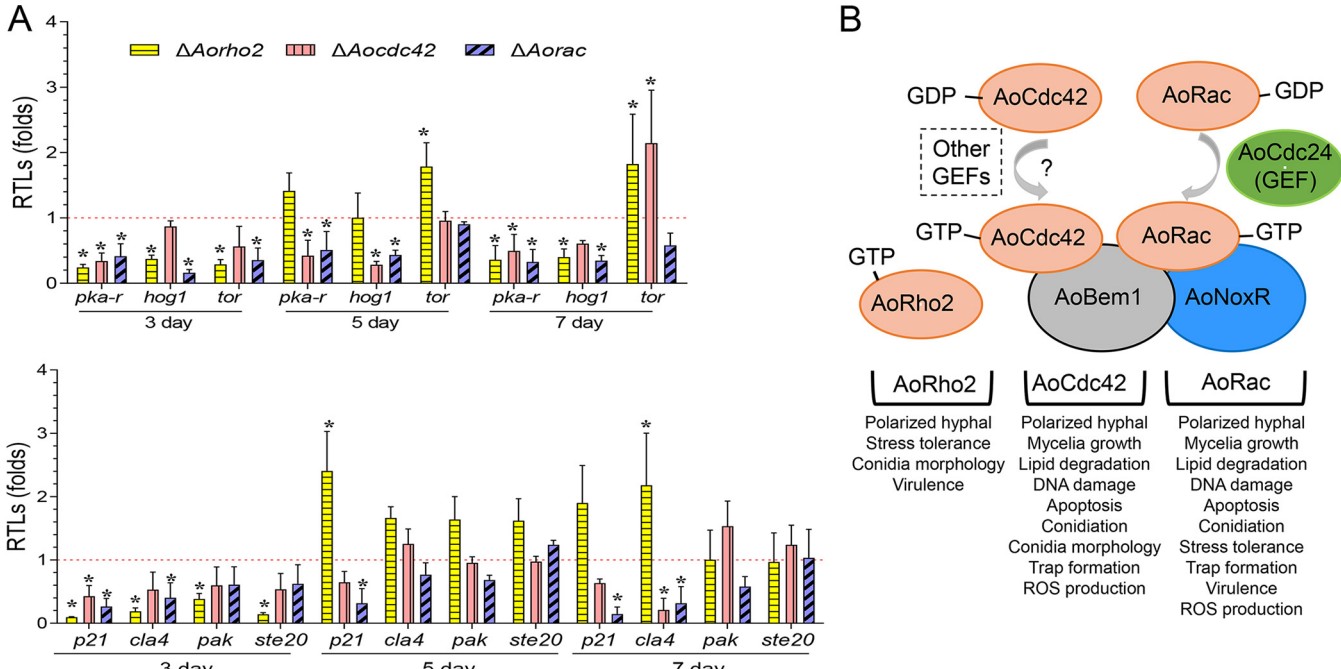

**FIG 10** Transcription levels of Rho GTPases' downstream effectors and summary of functions of Rho GTPases in *A. oligospora*. (A) RTLs of genes encoding downstream effectors of *AoRho2*, *AoCdc42*, and *AoRac*. The red line indicates the standard (with RTL = 1) for statistical analysis of the RTL of each gene in the respective deletion mutant versus that in the WT strain under a given condition. Bars and error bars represent mean values ± SD. *, significant difference between mutant and WT strain ($n = 3$ for each gene) (Tukey's HSD, $P < 0.05$). (B) Summary of functions of Rho GTPases and interactions between factors in the Nox complex and AoCdc42 and AoRac. AoNoxR, AoBem1, and AoRac interact with each other, and AoCdc42 and AoRac interact with AoBem1 and AoCdc24, respectively. AoCdc24 is a guanine nucleotide exchange factor (GEF) that activates AoRac; AoCdc42 may be activated by other GEFs. Activated AoRho2, AoCdc42, and AoRac play different roles in mycelium growth, lipid accumulation, DNA damage, sporulation, trap formation, pathogenicity, and stress response of *A. oligospora*.

the WT and mutant strains (Δ*Aorho2*, Δ*Aocdc42*, and Δ*Aorac*) were 53.6, 50.5, 36.8, and 11.4, respectively (Fig. 9D). Compared with the WT, the ROS production was reduced by 36% and 79% in the Δ*Aocdc42* and Δ*Aorac* mutants, respectively (Fig. 9E). In addition, the expression levels of two Nox-encoding genes, which we named *AonoxA* and *AonoxB*, were downregulated in the Δ*Aorac* mutant at the tested time points. In the Δ*Aocdc42* mutant, the *AonoxA* gene was downregulated on the 3rd and 5th days, whereas the *AonoxB* gene was downregulated on the 3rd and 7th days. In the Δ*Aorho2* mutant, *AonoxA* and *AonoxB* were upregulated at the tested time points (Fig. 9F).

**Deletion of Rho GTPase genes affected the expression of downstream effectors.** Reportedly, the activated Rho GTPases act on multiple effector molecules, such as p21-activated kinases (PAKs) and MAPKs that regulate numerous cellular processes (51). In this study, the expression levels of seven genes, encoding regulatory subunits of PKA (*pka-r*), Hog1-MAPK (*hog1*), mammalian target of rapamycin (*tor*), and four PAKs (*p21*, *cla4*, *pak*, and *ste20*), were compared between the WT and mutant strains. In the Δ*Aorho2* mutant, seven genes were downregulated on the 3rd day. In contrast, most genes were upregulated or no significant difference was observed on the 5th and 7th days, whereas *Pka-r* and *Hog1* were downregulated on the 7th day. In the Δ*Aocdc42* mutant, *Pka-r* and *P21*, and in the Δ*Aorac* mutant, *Pka-r*, *Hog1*, and *P21*, were significantly downregulated ($P < 0.05$) at the tested time points (Fig. 10A).

## DISCUSSION

In eukaryotes, gene transcription is essential for the realization of gene function (52). In this study, RNA-seq technology was used to generate a comprehensive transcriptome for *A. oligospora* induced with *C. elegans*. Our transcriptomic analysis demonstrated that nearly half of all genes were differentially expressed during the trap morphogenesis and infection process (12 to 48 h). AS occurred in 17.5% of the total

genes in *A. oligospora*. It has been shown that the AS rates in pathogenic fungi are higher than in nonpathogenic fungi; however, the rates vary among different pathogenic fungi (21). In *Ustilago maydis*, *M. grisea*, *B. bassiana*, and *C. neoformans*, the AS rates were estimated to be 3.6%, 7.9%, 19.9%, and 5.3%, respectively. At present, five main types of AS processes have been demonstrated, including SE, MXE, A5SS, A3SS, and RI in fungi (53). Here, we show that five types of splicing events occur in *A. oligospora* during trap development and infection; however, the relationship between AS events and pathogenicity needs further study. Analysis of the upregulated DEGs and AS-demonstrating genes during the trap development and infection process revealed that trap formation requires not only energy but also a series of complex cellular processes, such as metabolism of carbohydrates, amino acids, and lipids, biosynthesis, transport, and catabolism of secondary metabolites, peroxisome, phagosome cellular processes, cell cycle, and signal regulation. These results are consistent with those of our previous studies on genomic and proteomic analyses of the trap formation in *A. oligospora* (4, 5). In our previous studies, we have shown that signal pathways play a direct and primary role during the trap formation and lifestyle transition of *A. oligospora* (4, 5). In this study, several genes involved in MAPK and VEGF signaling pathways were upregulated at some time points during the infection process. In these pathways, *Aorac* was upregulated at three of the four tested time points (12, 36, and 48 h) and was significantly enriched to sphingolipid signaling, VEGF signaling, and phagosome pathways. Furthermore, *Aorho2* and *Aorac* demonstrated AS, and annotation with the PHI database revealed their association with loss or reduction of pathogenicity and lethality. These analyses implied that Rho GTPases play an essential role in the trap formation and nematode predation of *A. oligospora*.

Rho GTPases are highly conserved in fungi and participate in diverse critical processes, such as morphogenesis, development, and host infection (32, 54–58). Although Rho proteins are structurally similar and highly conserved across fungal species, increasing pieces of evidence suggest that their biological functions differ significantly (33, 41, 59, 60). To further elucidate the roles of Rho GTPases in NT fungi, we identified three orthologous Rho proteins (AoRho2, AoCdc42, and AoRac) in *A. oligospora* and investigated their roles in the physiology and pathogenicity of the fungus. The findings demonstrate that in *A. oligospora*, AoRho2 regulates mycelium branching, oxidation resistance, hyperosmotic resistance, cell wall synthesis, spore morphology, and extracellular proteolytic activity. This result is consistent with those observed in *Saccharomyces cerevisiae*, *Ashbya gossypii*, *A. niger*, and *N. crassa*—the *rho2* gene is required for branching and cell wall biosynthesis, and its deletion results in hyphal and colony morphologies similar to those of WT strains (28–31). Furthermore, we also showed that, like AoRac, AoCdc42 regulates mycelium branching, growth, and morphology, lipid accumulation, DNA damage and apoptosis, spore production, trap formation, nematocidal activity, and ROS production; however, AoRac was more prominent. Importantly, the Δ*Aorac* mutants lost the sporulation and trap formation abilities. The main functionality difference between AoCdc42 and AoRac was that the former regulates spore morphology, whereas the latter regulates the extracellular proteolytic activity and stress resistance. Concordant with these findings, the functions of Rac proteins have been shown to be prominent in *N. crassa*, *A. fumigatus*, and *A. niger* (29, 39, 40). Studies have also shown that Rac1 and Cdc42 are required for polar growth in *A. nidulans* (41), *N. crassa* (39), *M. oryzae* (32, 55), and *Fusarium graminearum* (58). In *M. oryzae*, Rac1, essential for conidiation, plays an important role in functional appressorium formation and is critical for virulence (32). Furthermore, in *F. graminearum*, deletion of the *Fgrac1*, *Fgcdc42*, and *Fgrho4* genes led to abnormal conidium morphology (58). Taken together, it can be inferred that AoRac and AoCdc42, more specifically AoRac, play a significant role in the regulation of phenotypic traits in *A. oligospora*.

Rac is a multifunctional small GTPase belonging to the Rho subgroup; one of its conserved functions is the activation of NADPH oxidases to produce ROS (50). NoxR is an essential regulatory component of the filamentous fungal NADPH oxidase (NoxA/

NoxB) complex, and polarity proteins Bem1 and Cdc24 are components of the NoxA/NoxB complex (61). Our analysis showed that AoNoxR, AoBem1, and AoRac could interact with each other, whereas AoCdc42 and AoRac interacted with AoBem1 and AoCdc24, respectively. In accordance with this, the interaction of RacA and Cdc42 with NoxR and BemA, respectively, is evident in the endophytic fungus *Epichloë festucae* (50). Similarly, in *U. maydis*, Cdc24 interacts with Rac (62). However, further studies are required to verify the interactions and their effects in *A. oligospora*. Moreover, our analysis showed that the intracellular ROS level was decreased in Δ*Aocdc42* and Δ*Aorac* mutants, and the reduction was more pronounced in Δ*Aorac* mutants. We have also shown that the expression levels of *AonoxA* and *AonoxB* were downregulated in the Δ*Aorac* (at all tested times) and Δ*Aocdc42* (at some time points) mutant strains. Reportedly, ROS act as a signal for cell differentiation to form traps, and knockout of *AonoxA* markedly reduces aerial hyphal growth, conidiation, ROS levels, trap formation, and insecticidal capacity (63). It was partially similar to the phenotypes observed in Δ*Aocdc42* and Δ*Aorac* mutants. Taken together, we conjectured that ROS levels between Δ*Aocdc42* and Δ*Aorac* mutant strains could be partly determined by their specific interactions with components of the Nox complex.

The Rho GTPases are highly conserved in eukaryotes. The core downstream effectors for Rho GTPases can be categorized into three distinct classes, namely, protein kinases, actin-binding proteins, and lipid-modifying enzymes (64). In *M. grisea*, *F. graminearum*, *Verticillium dahliae*, and *Claviceps purpurea*, the p21-activated kinase Cla4 homolog (Chm1 in *M. grisea*) functions downstream from Rac (32, 58, 65, 66). This study demonstrated that in the Δ*Aocdc42* and Δ*Aorac* mutant strains, the expression levels of genes encoding regulatory subunits of PKA, MAPK, and PAKs were downregulated ($P < 0.05$), and thus, PKA, P21, and Hog1 could be the downstream effectors of Rho GTPases in *A. oligospora*. However, we failed to obtain the *p21* deletion strain and demonstrate its interactions through Y2H. A recent study has shown that Hog1 is involved in the hyperosmotic resistance, conidiation, and trap formation of *A. oligospora* (45). Therefore, the pleiotropic effect of fungal Rho GTPases, presumably caused by the sharing of multiple downstream effectors, requires further analysis to elucidate the mechanism of how Rho GTPases activate various cellular events.

In summary, this study investigates the transcriptome profile of *A. oligospora* against *C. elegans* during the trap formation and infection process. Approximately half of the genome (5,752 DEGs) and a modest AS rate (~17.5%) are identified to be involved in the trap formation and infection process. Analysis of DEGs and genes in which AS occurred during these processes revealed that trap formation is an energy-consuming process and involves a series of complex cellular processes. More significantly, three Rho GTPases exerted differential effects on mycelium growth and morphology, lipid accumulation, DNA damage and apoptosis, spore production and morphology, trap formation, nematocidal activity, and ROS production in *A. oligospora* (Fig. 10B). Collectively, the current work provides an overview of gene expression and AS events in *A. oligospora* during trap development and nematode predation and highlights the important roles of the Rho family in fungal growth, development, trap formation, and lifestyle transition. The findings will help to elucidate the mechanisms of trap morphogenesis and to develop a more effective strategy for the biocontrol of pathogenic nematodes.

## MATERIALS AND METHODS

**Fungal strains and culture conditions.** The WT strain *Arthrobotrys oligospora* (strain ATCC 24927) and its derived mutants were cultured on PDA (200 g/liter potato, 20 g/liter dextrose, and 20 g/liter agar) plates at 28°C in the dark. *Caenorhabditis elegans* was incubated at 26°C on oatmeal agar medium (67).

**Sequence and phylogenetic analyses of AoRho2, AoCdc42, and AoRac.** The orthologous *rho2*, *rac*, and *cdc42* genes on the *A. oligospora* genome were identified using the sequences of the orthologs retrieved from the model fungi *S. cerevisiae*, *N. crassa*, and *A. nidulans* as queries. The amino acid sequences of AoRho2 (AOL_s00215g387), AoCdc42 (AOL_s00043g439), and AoRac (AOL_s00079g171) were downloaded from GenBank (http://www.ncbi.nlm.nih.gov/genbank/). The theoretical isoelectric point (pI) and molecular weight were calculated by the pI/Mw tool (http://web.expasy.org/compute_pi/). The conserved functional domains were predicted using InterProScan (http://www.ebi.ac.uk/Tools/pfa/iprscan/), and the amino acid sequences of Rho GTPases from different fungi were aligned using the

DNAman software package (version 5.2.2; LynnonBiosoft, St. Louis, Canada). A neighbor-joining tree was constructed using the Molecular Evolutionary Genetics Analysis (MEGA) version 6 software package (68). The three-dimensional structures of Rho GTPases were predicted by iterative threading assembly refinement (69).

**Knockout of *Aorho2*, *Aocdc42*, and *Aorac* genes.** For DNA isolation, mycelium was grown in a 500-ml Erlenmeyer flask (EF) containing 250 ml of PDB medium (200 g/liter potato and 20 g/liter dextrose) and incubated at 28°C and 180 rpm for 3 to 7 days. For gene deletion, a knockout cassette was constructed using the homologous recombination approach, as described previously (70). The primers used to amplify the flanking sequences of each gene are listed in Table S9. *A. oligospora* genomic DNA was used as a template. The hygromycin-resistance gene cassette (*hph*) was amplified using the primers hphF and hphR and the plasmid pCN44 as a template. The whole knockout cassette containing the *hph* gene with two homologous recombination arms was amplified using PCR with the primers 5f and 3r and was then used to transform into *A. oligospora* following a protoplast-based protocol (71). The positive transformants were further confirmed through PCR amplification using Yf and Yr primers for each gene (Table S9) and Southern blot analyses. The primer pairs Tf and Tr of each gene (Table S9) were used to generate the Southern hybridization probe using a PCR method. The restriction enzymes XhoI, BstEII, and HindIII were used to digest the genomic DNA of *A. oligospora* and the corresponding mutants (Δ*Aorho2*, Δ*Aocdc42*, and Δ*Aorac* mutants, respectively) for Southern blot analysis.

**Comparison of mycelial growth and morphology between the WT and mutants.** The WT and mutant strains were inoculated onto PDA, TYGA (10 g/liter tryptone, 5 g/liter yeast extract, 10 g/liter glucose, 5 g/liter molasses, and 20 g/liter agar), and TG (10 g/liter tryptone, 10 g/liter glucose, and 20 g/liter agar) media and incubated at 28°C for 7 days. Subsequently, their colony diameters and morphologies were recorded. The experiment was repeated three times for each strain.

**Comparison of stress resistance.** The WT and mutant strains were inoculated into TG medium supplemented with different concentrations of chemical stressors at 28°C for 7 days. The chemical stressors employed for the experiment were as follows: sorbitol (0.25, 0.5, and 1 M) and NaCl (0.1, 0.2, and 0.3 M) as osmotic stressors, SDS (0.01 to 0.03%) and Congo red (20 to 100 ng/ml) as cell wall-perturbing agents, and $H_2O_2$ (5 to 15 mM) and menadione (0.01 to 0.05 mM) as oxidative stressors. The RGI values of the strains were calculated as previously described (10).

**Comparison of conidial yield and germination rate.** The mutant and WT strains were inoculated in 250 ml EF containing 30 ml of CMY agar medium (20 g/liter Maizena, 20 g/liter agar, and 5 g/liter yeast extract), followed by incubation at 28°C for 14 days. The conidia of WT and mutants were collected from CMY medium by scraping them off with a glass spatula into sterile distilled $H_2O$. The number of conidia was determined using a hemocytometer (72, 73).

For conidial germination tests, approximately $1 \times 10^5$ conidia were inoculated into 30 ml Vogel's medium (minimal) (20 ml/liter Vogel's salts and 15 g/liter sucrose) in a 250-ml EF and incubated at 28°C and 180 rpm for 4, 8, and 12 h to assess conidial germination rates (14). The experiments were performed in triplicate.

**Bioassay against the nematode *C. elegans*.** Approximately $1 \times 10^4$ conidia were cultivated on WA plates. After 3 days of incubation at 28°C, 200 nematodes (*C. elegans*) were added to the center of each plate. Since Δ*Aorac* mutants did not produce conidia, mycelium blocks were cultured in CMY agar medium at 28°C, and 200 nematodes were added to induce trap formation when the colony size was uniform. The numbers of traps and captured nematodes were counted under a light microscope (BX51; Olympus, Tokyo, Japan) at 12-h time intervals.

**Determination of proteolytic activity.** The fungal strains were inoculated into PDB medium and incubated at 28°C and 180 rpm for 7 days. The fermented liquid was collected, and the proteolytic activity was determined on casein plates (74). Moreover, the protease activity was quantified using a caseinolytic method described previously (74, 75). One unit (U) of protease activity was defined as the amount of enzyme that hydrolyzed the substrate and produced 1 $\mu$g of tyrosine in 1 min under the specific assay conditions.

**Transcriptome sequencing and analysis.** The *A. oligospora* strains (WT and mutant) were cultured in PDA medium for 3 days, and then they were transferred to WA medium with low nutrition. Subsequently, the mycelia of fungal strains were treated with *C. elegans* from 0 to 48 h, and 15 samples were obtained for further analysis. Mycelial samples were sent to Wuhan Seqhealth Technology Co. Ltd. for transcriptome sequencing and analysis. GO and KEGG analyses were performed using the OmicShare tools, a free online platform for data analysis (https://www.omicshare.com/tools). AS analysis was performed using the free online platform of Majorbio Cloud Platform (http://www.majorbio.com).

**Morphological observation, nuclear staining, and TUNEL analysis.** For the observation of mycelial and conidial morphology, the WT and mutant strains were stained with 10 $\mu$g/ml calcofluor white (Sigma-Aldrich, St. Louis, MO, USA), and the cell nuclei of mycelia and conidia were visualized by staining the cells with 10 $\mu$g/ml DAPI for 30 min (76). The samples were observed under a confocal laser scanning microscope (Leica, Mannheim, Germany) or light microscopy.

DNA fragmentation and cell apoptosis were determined via the TUNEL assay, followed by staining the nucleus with 10 $\mu$g/ml PI for 10 min. TUNEL assay using the one-step TUNEL apoptosis detection kit (Beyotime, Jiangsu, China) was performed according to the manufacturer's protocol. The cells were then observed under a confocal laser scanning microscope, and the fluorescence intensity was estimated using the ImageJ software (https://imagej.net/Welcome).

**Analysis of the ROS levels and lipid accumulation.** ROS and lipid droplets were stained with 10 $\mu$g/ml DHE (Beyotime, Jiangsu, China) and BODIPY (Sigma-Aldrich, St. Louis, MO, USA) for 10 min

each. The samples were observed under a confocal laser scanning microscope. The fluorescence intensities were determined using the ImageJ software (https://imagej.net/Welcome).

**Y2H assay.** Yeast two-hybrid assays using pGADT7 or pGBKT7 (Clontech, Kusatsu, Japan)-based constructs were performed according to the manufacturer's instructions (Matchmaker two-hybrid system 3; Clontech). Yeast strain AH109 was transformed with prey (pGADT7 derivatives) and bait (pGBKT7 derivatives) vectors. Transformants were plated on synthetic dropout medium lacking leucine and tryptophan (double dropout [DDO]), or leucine, tryptophan, histidine, and adenine (quadruple dropout [QDO]). Growth on the latter indicates an interaction between bait and prey.

**Preparation of RNA and RT-qPCR analysis.** Total RNA was isolated from the samples using an RNA extraction kit (Axygen, Jiangsu, China) and reverse transcribed into cDNA with a PrimeScriptH$^{RT}$ reagent kit (with genomic DNA [gDNA]; TaKaRa, Kusatsu, Japan). The cDNA was used as a template to determine the mRNA expression of candidate genes associated with phenotypes like conidiation and stress resistance by performing RT-qPCR with specific paired primers and the LightCycler 480 with SYBR green I master mix (Roche, Basel, Switzerland) (Table S9). $\beta$-Tubulin was used as an internal standard. All RT-qPCR experiments were performed in triplicates. The relative transcript level (RTL) of each gene was calculated as the ratio of the cycle threshold ($C_T$) value for the gene in the mutant strain to that in the WT strain using the $2^{-\Delta\Delta CT}$ method (77).

**Validation of AS events.** RT-PCR was used to validate the alternative splicing events. Total RNA extraction and cDNA synthesis were performed as described above. Primers used in this study are listed in Table S9.

**Statistical analyses.** The data from each experiment were expressed as mean values $\pm$ standard deviations (SD). One-way analysis of variance followed by an honestly significant difference (HSD) test was used for statistical analyses, and a $P$ value of $< 0.05$ was considered significant. All statistical analyses were performed using the GraphPad Prism software version 5.00 for Windows (GraphPad Software, San Diego, CA, USA).

**Data availability.** All data generated or analyzed during this study are included in the published paper and associated supplemental files. All transcriptomic data are reported in supplemental files of this paper. The raw sequence has been deposited to GEO under accession number GSE192443.

## SUPPLEMENTAL MATERIAL

Supplemental material is available online only.
**SUPPLEMENTAL FILE 1**, PDF file, 2.4 MB.
**SUPPLEMENTAL FILE 2**, XLSX file, 0.01 MB.
**SUPPLEMENTAL FILE 3**, XLSX file, 0.02 MB.
**SUPPLEMENTAL FILE 4**, XLSX file, 4.5 MB.
**SUPPLEMENTAL FILE 5**, XLSX file, 0.02 MB.
**SUPPLEMENTAL FILE 6**, XLSX file, 0.04 MB.
**SUPPLEMENTAL FILE 7**, XLSX file, 3.2 MB.
**SUPPLEMENTAL FILE 8**, XLSX file, 0.03 MB.
**SUPPLEMENTAL FILE 9**, XLSX file, 0.1 MB.
**SUPPLEMENTAL FILE 10**, XLSX file, 0.02 MB.

## ACKNOWLEDGMENTS

We thank Guo Yingqi (Kunming Institute of Zoology, Chinese Academy of Sciences) for her help in obtaining and analyzing transmission electron microscopy images.

Funding for this study was provided by the National Natural Science Foundation of China (grants no. 31960556 and U1402265) and the Applied Basic Research Foundation of Yunnan Province (grant no. 202001BB050004).

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
