## [Reviewer comments · Microbiology Spectrum]

Microbiology Spectrum

Transcriptomic analysis reveals that Rho GTPases regulate trap development and lifestyle transition of the nematode-trapping fungus *Arthrobotrys oligospora*

Jinkui Yang, Le Yang, Xuemei Li, Na Bai, Xuwei Yang, and Ke-Qin Zhang

Corresponding Author(s): Jinkui Yang, Yunnan University, China

Review Timeline:

Submission Date:

November 22, 2021

Accepted:

December 2, 2021

Editor: Christina Cuomo

Reviewer(s): The reviewers have opted to remain anonymous.

Transaction Report:

DOI: <https://doi.org/10.1128/spectrum.01759-21>

December 2, 2021

Prof. Jinkui Yang
Yunnan University, China
No. 2 of Cuihubeilu
Kunming, Yunnan province 650091
China

Re: Spectrum01759-21 (Transcriptomic analysis reveals that Rho GTPases regulate trap development and lifestyle transition of the nematode-trapping fungus *Arthrobotrys oligospora*)

Dear Prof. Jinkui Yang:

Your manuscript has been accepted, and I am forwarding it to the ASM Journals Department for publication. You will be notified when your proofs are ready to be viewed.

Sincerely,

Christina Cuomo
Editor, Microbiology Spectrum

Journals Department
Supplemental Material 8: Accept
Supplemental Material 1: Accept
Supplemental Material 3: Accept
Supplemental Material 4: Accept
Supplemental Material 2: Accept
Supplemental Material 6: Accept
Supplemental Material 5: Accept
Supplemental Material 10: Accept
Supplemental Material 7: Accept
Supplemental Material 9: Accept